# CD38, CD157, and RAGE as Molecular Determinants for Social Behavior

**DOI:** 10.3390/cells9010062

**Published:** 2019-12-25

**Authors:** Haruhiro Higashida, Minako Hashii, Yukie Tanaka, Shigeru Matsukawa, Yoshihiro Higuchi, Ryosuke Gabata, Makoto Tsubomoto, Noriko Seishima, Mitsuyo Teramachi, Taiki Kamijima, Tsuyoshi Hattori, Osamu Hori, Chiharu Tsuji, Stanislav M. Cherepanov, Anna A. Shabalova, Maria Gerasimenko, Kana Minami, Shigeru Yokoyama, Sei-ichi Munesue, Ai Harashima, Yasuhiko Yamamoto, Alla B. Salmina, Olga Lopatina

**Affiliations:** 1Department of Basic Research on Social Recognition and Memory, Research Center for Child Mental Development, Kanazawa University, Kanazawa 920-8640, Japan; minahashii@gmail.com (M.H.); rgabata@gmail.com (R.G.); makoto_tsubomoto@yahoo.co.jp (M.T.); nseishima@med.kanazawa-u.ac.jp (N.S.); mitsuyo.teramachi@gmail.com (M.T.); kamiji0029@yahoo.co.jp (T.K.); ctsuji@med.kanazawa-u.ac.jp (C.T.); stas4476@mail.ru (S.M.C.); ashabalova@me.com (A.A.S.); mgera_08@mail.ru (M.G.); minami-k@staff.kanazawa-u.ac.jp (K.M.); shigeruy@med.kanazawa-u.ac.jp (S.Y.); 2Laboratory of Social Brain Study, Research Institute of Molecular Medicine and Pathobiochemistry, Krasnoyarsk State Medical University named after Prof. V.F. Voino-Yasenetsky, Krasnoyarsk 660022, Russia; allasalmina@mail.ru (A.B.S.);; 3Division of Molecular Genetics and Clinical Research, National Hospital Organization Nanao Hospital, Nanao 926-0841, Japan; 4Molecular Biology and Chemistry, Faculty of Medical Science, University of Fukui, Fukui 910-1193, Japan; ytanaka@u-fukui.ac.jp; 5Life Science Research Laboratory, University of Fukui, Fukui 910-1193, Japan; ma-tsu-ka@hotmail.co.jp; 6Molecular Pharmacology, Suzuka University of Medical Science, Suzuka 513-0816, Japan; higuchiyjp@yahoo.co.jp; 7Department of Neuroanatomy, Kanazawa University Graduate School of Medical Sciences, Kanazawa 920-8640, Japan; thattori@staff.kanazawa-u.ac.jp (T.H.); osamuh3@staff.kanazawa-u.ac.jp (O.H.); 8Department of Biochemistry and Molecular Vascular Biology, Kanazawa University Graduate School of Medical Sciences, Kanazawa 920-8640, Japan; smunesue@med.kanazawa-u.ac.jp (S.-i.M.); aharashima@staff.kanazawa-u.ac.jp (A.H.); yasuyama@med.kanazawa-u.ac.jp (Y.Y.)

**Keywords:** CD38, CD157, RAGE, NAD, cyclic ADP-ribose (cADPR), social memory, oxytocin transporter, autism, anxiety

## Abstract

Recent studies provide evidence to support that cluster of differentiation 38 (CD38) and CD157 meaningfully act in the brain as neuroregulators. They primarily affect social behaviors. Social behaviors are impaired in *Cd38* and *Cd157* knockout mice. Single-nucleotide polymorphisms of the *CD38* and *CD157/BST1* genes are associated with multiple neurological and psychiatric conditions, including autism spectrum disorder, Parkinson’s disease, and schizophrenia. In addition, both antigens are related to infectious and immunoregulational processes. The most important clues to demonstrate how these molecules play a role in the brain are oxytocin (OT) and the OT system. OT is axo-dendritically secreted into the brain from OT-containing neurons and causes activation of OT receptors mainly on hypothalamic neurons. Here, we overview the CD38/CD157-dependent OT release mechanism as the initiation step for social behavior. The receptor for advanced glycation end-products (RAGE) is a newly identified molecule as an OT binding protein and serves as a transporter of OT to the brain, crossing over the blood–brain barrier, resulting in the regulation of brain OT levels. We point out new roles of CD38 and CD157 during neuronal development and aging in relation to nicotinamide adenine dinucleotide^+^ levels in embryonic and adult nervous systems. Finally, we discuss how CD38, CD157, and RAGE are crucial for social recognition and behavior in daily life.

## 1. Introduction

Accumulating evidence indicates that immune-related molecules tend to be involved in the neuronal system. Unexpectedly a number of immune molecules were reported to possess neuronal functions [1]. Among them, cluster of differentiation 38 (CD38) is a transmembrane glycoprotein and has ADP-ribosyl cyclase activities and functions in cell activation on many hematopoietic, plasma, and B and T lymphocytes [2,3,4]. Another example is CD157, first identified as bone marrow stromal cell antigen-1 (BST-1) from rheumatoid patients [5,6,7]. CD157 is a glycosylphosphatidylinositol-anchored molecule that facilitates pre-B-cell growth [6]. The deduced amino-acid sequence of CD157 exhibits 33% similarity with CD38 [3,4].

The first study to report physiological functions of CD38 in the brain was published by Jin et al. in 2007, in that CD38 has an important role in oxytocin (OT) secretion in the hypothalamus and in regulating of social memory and social interactions [8,9,10]. The mechanism underlying the regulation of OT release from OT-containing neurons involves increases in intracellular free Ca^2+^ concentrations by mobilization of Ca^2+^ through ryanodine receptors type II or III [3,4,11,12] from intracellular Ca^2+^ pools by cyclic ADP-ribose (cADPR).

An exciting study was reported in 2009 that genome-wide association studies (GWASs) identified single-nucleotide polymorphisms (SNPs) in the *CD157/BST1* gene as risk factors for Parkinson’s disease in the Japanese cohort [13], followed by confirmation in different populations in subsequent reports [14]. These human GWASs suggest some kind of roles of CD157 in the brain. However, no evidence on the neuronal role of CD157 is available for the past five years, since the discovery of the SNPs. Interestingly, the results obtained from gene deletion of *Cd157* mice showed merely motor dysfunction but social behavioral impairments, including apathy-, anxiety-, and depression-like behaviors [15,16]. Of course, social deficits are also important symptoms of Parkinson’s disease. These *Cd157* knockout (KO) mice are expected to analyze such non-motor impairments in order to improve quality of life.

Now, immune molecules, CD38 and CD157, are the immuno-neuronal molecules which also regulate neuronal function. Because functional roles of CD157 in the intestinal tissue were reported in 2012 [17], this molecule is the immuno-entero-neuronal molecule [6,18]. In this article, we mainly mention various characteristics of CD38 and CD157 in relation to brain function, behavior, and impaired behavior in KO mice which resemble the developmental disorders such as autism spectrum disorder (ASD) [10,12,18]. We make clear the similarity and dissimilarity between CD38 and CD157, which sheds light on future questions to be asked for the function of such molecules.

## 2. Genes and Single-Nucleotide Polymorphisms

The *CD38* and *CD157/BST1* genes are located on the subregion of the human chromosome 4p15 as a next neighbor. The genescape is well documented [3,4,5,6,7,19,20,21].

For an association study of *CD38* and ASD, 10 intronic SNPs of *CD38* were examined in a case–control study in a Japanese population. No significant association with ASD was identified in these SNPs [21]. Furthermore, when performed in the United States (US) ASD DNA cohort (selected Caucasian 252 trios in the Autism Genetic Resource Exchange (AGRE) samples), none of the selected SNPs showed significant associations [21]. However, if focused only in the US high functioning autism subgroup, SNPs of 104 trios in our AGRE revealed association in rs6449197 (*p* = 0.040) and rs3796863 (*p* = 0.005). Unfortunately, no association was detected in Japanese high functioning autism trio cases (*p* = 0.228). With respect to one exonic SNP, rs1800561 (4693C > T), some Asian ASD patients and controls possess arginine (dominant) and/or tryptophan at the 140th amino acid of CD38. Although there was no clear association in the SNP, ASD probands carrying tryptophan CD38, instead of arginine, were segregated in three Japanese families examined [21].

These initial SNPs analyzed for *CD38* including rs3796863 were extended to infants’ attention to social eye cues [22] and replicated in ASD cases [23,24,25,26,27,28]. Most recently, in association studies of *CD38*, meaningful association was found in anorexia nervosa [29] and in suicidal ideation [26,30].

GWASs and meta-analyses for Parkinson’s disease identified intronic SNPs in the *CD157/BST1* gene as new susceptibility loci in Asian and European populations [13,14,31]. However, it was pointed out that environmental factors may also contribute to the real pathogenic role of *CD157* SNPs in Parkinson’s disease [32].

Yokoyama et al. found associations between ASD and three SNPs of *CD157* (rs4301112, rs28532698, and rs10001565) [33]. These three SNPs have chromosomal locations (from chr4:15717226 to chr14:15722573) un-identical to those associated with the region in Parkinson’s disease (chr14:15725766 to chr4:15737937) [13,14]. These studies revealed that some SNPs in *CD157* may be risk factors for both ASD and Parkinson’s disease and those in *CD38* may be risk factors for several psychiatric disorders.

## 3. Messenger RNA (mRNA) Expression Patterns during Development

The rodent brain *Cd38* mRNA levels sharply increased on postnatal days 7–14. In rats, the level reached a 100-fold increase in adult and, in mice, a 25-fold increase was seen from no or little level at embryo or neonates [18,34]. In contrast, the time course of *Cd157* mRNA is opposite to that of *Cd38.* The brain *Cd157* mRNA levels decreased 7–14 days postnatally from the relatively higher levels in the embryonic days [18]. However, such a relationship is yet to be studied in humans. *Cd38* mRNA was expressed in the four mouse brain sub-regions (hypothalamus, cerebellum, striatum, and cerebrum) [9], with the highest density in the hypothalamus (Supplementary Figure 11 of Jin et al. [9]).

The levels of *Cd157* mRNA expression in the brain regions in adult male C57BL/6 mice were very low [18]. These mRNA expression studies showed the quite distinct expression profiles of *Cd38* and *Cd157*.

## 4. Proteins in the Brain

The high level of mRNA of *CD38* expression was measured in the human hypothalamus [21]. CD38 immunoreactivity was detected in many cells in the paraventricular nucleus of the hypothalamus, while much lower CD38 expression levels were observed in the insular cortex [21]. While there was co-expression of OT and CD38 in the hypothalamic paraventricular nucleus (PVN), there was little or no detectable OT in other parts of the cortex.

There were a number of studies reporting CD38 expression in glial cells, including astrocytes, microglia, and neurons [35,36,37,38,39,40,41,42,43]. Yamada et al. [40] reported by using ultrastructural analysis that CD38 immunoreactivity was detected in a subset of pyramidal neurons and astrocytes, but not in microglia or oligodendrocytes under physiological conditions in rat. Neuron–glia crosstalk also showed that astrocytes increased CD38 expression after the glutamate release from neurons [44]. In brain ischemia, CD38 would be predominantly expected in astrocytes to support neuronal survival by transferring extracellular mitochondria from astrocytes to neurons [45].

Furthermore, Hattori et al. [38] demonstrated that CD38 is already expressed in the developing brain between postnatal day 14 and day 28. In situ hybridization and flow cytometrical analysis revealed that CD38 is expressed in astrocytes at these times [38]. In association with this, astrocytic development is delayed in *Cd38* KO mice; subsequently delayed differentiation of oligodendrocytes at postnatal periods was obvious, which was replicated in the culture system. It is reasonable to conclude that astrocytic CD38 regulates the development of astrocytes in a cell-autonomous way and the differentiation of oligodendrocytes in a non-cell-autonomous manner [39]. Further to this, connexin43 in astrocytes plays a promotive role for CD38-mediated oligodendrocyte differentiation. Finally, increased levels of nicotinamide adenine dinucleotide^+^ (NAD^+^), caused by *CD38* gene deficiency, suppressed astrocytic connexin43 expression and oligodendrocyte differentiation. CD38 is a positive regulator of astrocyte and oligodendrocyte development [39].

CD38 expression was reported in glial cells, such as astrocytes and microglia [37,40]. Astrocytic CD38 may regulate the maturation of astrocytes and/or differentiation of oligodendrocyte precursor cells into oligodendrocytes by metabolizing NAD^+^ in the brain in physiological conditions [38].

In pathological conditions, CD38 seems to be involved in the activation of microglia and astrocytes in mouse models of glioma or traumatic brain injury, and human immunodeficiency virus (HIV)-infected brains [46,47,48]. The role of CD38 in demyelination was recently studied in cuprizone-induced demyelination in mice, which is evident by oligodendrocyte-specific apoptosis, followed by intensive glial activation, demyelination, and repopulation of oligodendrocytes [39]. CD38 was upregulated in astrocytes and microglia after cuprizone application. *CD38* deficiency did not affect the initial decrease of the number of oligodendrocytes, while it attenuated cuprizone-induced demyelination and neurodegeneration [39]. The clearance of the degraded myelin and oligodendrocyte repopulation could be reduced in *Cd38* KO mice. This was associated with reduced levels of glial activation and inflammatory responses, such as phagocytosis through the increased level of NAD^+^ in CD38-deleted conditions. CD38 and NAD^+^ in the glial cells may play an important role in the demyelination and subsequent oligodendrocyte remodeling by the modified glial activity and neuroinflammation.

## 5. Immunohistochemistry of CD157 in Neural Stem Cells

Immunofluorescence images of CD157 were seen in the E17 embryo hypothalamus [18]. CD157 staining was seen in the cytoplasm or at the cell surface of a subset of Nestin-positive cells in the ventricular and subventricular zones near the third ventricle (Figure 1). However, this colocalization seems to be limited to the embryonic stem cells in the mouse. The question remains as to whether or not CD157 is present in adult mouse brain stem cells.

Recently, Wu et al. showed that CD157 is expressed in CD45^–^, CD54^+^, CD157^+^ lung stem/progenitor cells [49]. Yilmaz et al. showed that fasting in mice results in activating the mammalian target of rapamycin1 (mTOR1) pathway and induction of CD157 in Paneth cells in the adult intestine [17,50]. cADPR produced by CD157 can inhibit the differentiation of intestinal stem cells into acinar cells, thus facilitating self-renewal of intestinal stem cells [17]. These results suggest that CD157 functions in stem cells in two other organs.

Very recently, CD157 was discovered in an endothelial stem cell population required for vascular regeneration and tissue maintenance [51], as well as in the mammary gland [52]. cADPR may function in these stem cells, as shown in Paneth cells [17]. In the nervous system, however, CD157 plays a role in neuronal migration during neural stem cell growth and neurogenesis. It was indicated that CD157 binds to members of the integrin family [53,54]. Therefore, it would be interesting to examine whether cADPR facilitates the self-renewal of stem cells and forms a complex of CD157 and integrin.

In addition, it was found that CD157 is resident of endothelial cells and a hallmark [51,55]. CD157^+^, CD200^+^ endothelial cells play a role in a stem cell population with self-renewal capacity as determined blood vessels. Such endothelial stem cells can differentiate into CD157^−^, CD200^−^ endothelial cells through CD157^+^, CD200^−^ (progenitor) cells [55]. Thus, it is interesting to find such markers in neural stem cell development.

## 6. Binding of CD38 to Calcium Calmodulin-Dependent Kinase

CD38 produces Ca^2+^-mobilizing second messengers via its catalytic site [3]. With respect to molecular mechanisms of enzyme activity modulation, we recently detected CD38-interacting proteins by proteomic analysis using two-dimensional gel electrophoresis followed by mass spectrometry [56]. We identified the β-isoform of Ca^2+^-calmodulin-dependent protein kinase II (CAMKII) as the human CD38 (hCD38)-binding protein from HEK-293 cells overexpressing hCD38 (Figure 2).

Pharmacological methods confirmed the functional interaction of CAMKIIβ and CD38 [56]. Briefly, in *human influenza hemagglutinin* (*HA)-CD38* complementary DNA (cDNA)-transfected human embryonic kidney (HEK) culture cells, the wound open space surrounded with a healing border became small after 24 h. The transfected cells treated with an inhibitor of CAMKII, KN-62, did not close the open space without wound healing (Figure 3). Quantitatively, *CD38* cDNA-transfected samples treated with KN-62 had significantly less migratory activity compared with those without KN, suggesting a possible interaction between CD38 and CAMKII.

For the point of targets, CAMKII activated by increased cytosolic Ca^2+^ functions as a multifunctional serine/threonine kinase to activate downstream signaling pathways [57], such as the activation of ryanodine receptors by phosphorylation [58]. Another candidate target of activated CAMKII is actin dynamics. The cytoskeleton protein actin (F-actin) binds to CAMKII [59], and this binding is specific to the β-isoform [57]. It was shown that CaMKIIβ binds to monomeric actin to reduce actin polymerization. When Ca^2+^ binds to CaMKIIβ, the CaMKIIβ–actin interaction is antagonized, and actin polymerization is facilitated [60]. Therefore, one possibility may be that CAMKIIβ and CD38 co-operate for regulation of actin dynamics characterized by such polymerization–depolymerization reactions. The efficient activation of CAMKIIβ by bound CD38 causes release of monomeric actin sequestration and facilitates actin polymerization, thereby facilitating cell migration, because the migration process requires actin polymerization [61]. In addition, one observation that CD38 proteins are associated with actin [62] supports the above idea of CD38-dependent modulation of actin dynamics.

The molecular weight of CD38 bound to CAMKIIβ in HEK cells is lower than that of glycosylated CD38 [56]. It was reported that type III CD38 is not usually glycosylated [63]. This leads to speculation of the possibility that type III CD38 and CAMKIIβ form a complex, rather than the type II CD38, as schematized in Figure 4. Physically coupled CD38 and CAMKII may functionally activate downstream cellular signals efficiently, as demonstrated above in cell motility. Most recently, it was revealed that the type III CD38 interacts with integrin binding protein 1 [64] or transferrin, CD71 [65].

## 7. Enzymatic Activities of CD38 and CD157

It was established that CD38 and CD157 belong to an ADP-ribosyl cyclase family, which elicits the formation of cADPR and/or ADP-ribose from β-NAD^+^ [3,4,5,6,7,9,17,18,38]. The ADP-ribosyl cyclase activity of CD157 is much weaker than that of CD38 [3,18]. The product of base exchange is nicotinic acid adenine dinucleotide phosphate (NAADP) [3,66,67]. NAADP also has Ca^2+^ mobilization activity from different Ca^2+^ pools [6]. Kim and his colleagues examined whether CD157 has base exchange activity, through independently transfecting with mouse *Cd157* or *Cd38* genes, in which both proteins were well expressed [18]. Firstly, the ADP-ribosyl cyclase activity of *Cd157*-transfected HEK cells was lower than that of *Cd38*-transfected HEK cells. *Cd157*-transformed cells showed no or little NAADP synthetic activity, with no discernable differences from mock-transfected cells, suggesting that CD157 has little or no base exchange activity. In sharp contrast, *Cd38*-transfected HEK cells displayed higher NAADP synthetic activity compared to control cells [18], suggesting that the functional role of CD157 likely stems from the production of cADPR rather than NAADP. However, a question still remains whether CD157 in the nervous system mediates its functions in stem cells through cADPR but not NAADP. To consider this, the crucial role of both cADPR and CD157 in intestinal Paneth cells will give us a hint [17].

## 8. Contribution of TRPM2 on Ca Signaling

It was reported that NAD^+^ metabolites target several ion channels, including transient receptor potential melastatin 2 (TRPM2), which is a member of the warmth-sensing family that plays an important role in thermoregulation [68,69]. Activation of TRPM2 non-specific cation channels results in Ca^2+^ influx in response to the temperature range from 34 to 40 °C [68]. Therefore, initially, we thought that OT release could be facilitated by activation of TRPM2 channels. Extracellular application of 100 μM cADPR increases intracellular free calcium concentrations, together with simultaneous temperature elevation by 2 °C from 35 °C in the presence of extracellular Ca^2+^ in a single cultured cell of the anterior hypothalamus of the mouse [36]. Its temperature sensitivity resembles that of TRPM2 channels as a warm sensor, in that Ca^2+^ influx through TRPM2 occurs at temperatures of the body temperature range from 34 °C to 39 °C [68,69].

On the contrary, β-NAD^+^ showed a constant effect during the sustained phases required for conversion of β-NAD^+^ to cADPR; thus, its effect seems to be identical to cADPR in the late phase. The initial free intracellular Ca^2+^ concentration rise is more sensitive to Ca^2+^ influx due to thermal stimulation, while ADPR and the late sustained phase are dependent on cADPR-involved Ca^2+^ mobilization from the ryanodine-sensitive Ca^2+^ pool. Very recently, an endogenous cADPR analogue, 2′-deoxyadenosine 5′-diphosphoribose, was documented and demonstrated to be very powerful [70]. It is of interest to test it in OT release in the hypothalamus.

Incubation with 100 μM cADPR alone induced minor increases in concentrations of OT in the culture medium of a single hypothalamus from the group-housed wild-type mice, but the OT concentration increased with additional thermal stimulation. These transient and accumulated responses with heat and cADPR suggest that heat and cADPR have distinct effects on OT release. Further confirmation was that the cADPR- and heat-induced OT concentration increase was not observed in *Cd38* KO mice. Overall, the results suggest that the cADPR- and heat-induced release of OT depends on CD38 and TRPM2. They provoke cADPR-producing enzyme activity and Ca^2+^ influx through cation channels in the mouse hypothalamus that experienced social stress during group-housing [36,71]. Thus, these KO experiments revealed that both CD38 and TRPM2 are involved in the cADPR- and heat-induced facilitation of OT release in vivo. Furthermore, now, cooperative roles of TRPM2 and CD38 were reported in natural killer cells [72], mesenchymal stem cells [73], pancreatic beta cells [74], neutrophil granulocytes [75], and inflammatory monocytes [76]

## 9. Social Behavior in Knockout Mice

Severe anxiety-like and depression-related behaviors are displayed in *Cd157* KO male mice in the light–dark transition test, known as a standard anxiety-related behavioral test [16,77,78]. The transition from a light to a dark arena was shorter in the *Cd157* KO mice, compared with the controls. However, the same test should be performed even in *Cd38* KO mice. While the entry frequency was different, probably because of the different genetic backgrounds of the two strains, no identical behavioral differences were found between *Cd38* KO and wild-type ICR mice [79]. These findings suggest that anxiety induced by novel environment is a distinct feature of *Cd157* KO mice, but not *Cd38* KO mice, or recently used receptor for advanced glycation end-products (*RAGE*) KO mice [16,71,77,80,81] (Figure 5).

In the open field test, the time spent in the inside zone was shorter for *Cd157* KO mice than wild-type controls. Interestingly, this phenotype seemed to be recovered by a single intraperitoneal administration of OT with duration in the inside zone and immobility time increasing significantly [78].

*Cd157* KO mice showed depression-like behaviors in the forced swimming and tail suspension tests [16]. Anxiety-like behaviors were evident in the light–dark transition test and in the elevated plus maze test. Such impaired social behaviors in the social preference test in part seem to resemble psychiatric symptoms [18].

## 10. Differential Roles of CD38 and CD157 in Social Behavior

The role of *CD38* in OT secretion into the brain is already established [9,10,11,21,22]. CD38 mediates multiple reactions such as ADPR and cADPR production, TRMP2 and extracellular signal-regulated kinase (ERK1/2) activation, Ca^2+^-mobilization, and OT release [68]. Additionally, CD38 acts in OT release by activating molecular cascades of OT autoregulation [9,10]. In contrast, CD157 can bind with the serotonin transporter and integrin-β3 and invoke multiple circuits to control anxiety- and depression-like behaviors [16,53,54]. CD157 may play a role in cADPR-induced OT release, which is not completely clear yet (Figure 5). The deficiency of CD157 subsequently leads to aberrant behaviors, such as increased anxiety. A decrease in volume of the amygdala [16], a key constituent of the social brain, might be generated by a loss of CD157 in the neural stem cells during the embryonic developmental days. This question should be further studied.

## 11. Autism Spectrum Disorder

Hyperthermia is known to be induced by social stress [71,82,83,84]. The social stress condition induces a much more stressful effect on a subset of mice subordinate in social status than dominant in social hierarchy. Thus, in the subordinate mouse group, the release of more OT allowed them to recover from stress, which resulted in becoming tolerant to greater stress. When more stress was experienced, more OT could be released subsequently to achieve a balance, which is very medically congruent [85] (Figure 5).

There are several unique studies of fever in ASD patients. Some ASD children exhibit amelioration in their characteristic ASD behaviors during febrile incidents. Possibly, the regression of fever may be associated with the onset of ASD [86,87]. Febrigenesis and the behavioral state changes associated with fever in ASD depend on the selective normalization [87]. Fever may enhance the release of OT, which can subsequently reduce abnormal ASD behavior, because external or nasal administration of OT was proven to improve aberrant behavior in rodents and humans [9,87,88] (Figure 5).

The behavioral impairments in *Cd157* KO mice were recovered by OT, probably because OT directly targets the intracellular signaling networks in the social brain related to CD157, independently differing from CD38 [71]. This suggests that OT can be used in future for the treatment of social withdrawal in psychiatric disorders. Whether we can extend the results observed in the mouse to human behavioral recuperation is of interest, especially since there are reports regarding the effectiveness of OT for impaired social interaction as a core symptom of ASD [89,90,91,92,93].

The report by Zhong et al. [36] presented a new neuroendocrinological concept that fever may increase the release of OT to reduce abnormal autistic behavior. The reason for this is that external administration of OT would improve aberrant behavior in rodents and in humans (Figure 5 and Figure 6).

Exogenous OT in humans induces behavioral effects, particularly in social deficit-related psychiatric disorders such as ASD [8,10,12,94]. Several clinical trials reported the neuropharmacological effects of peripherally administered OT in a variety of conditions. The potential for peripherally administered OT to act centrally is supported by measures of OT in the brain of wild-type and *RAGE* KO mice or OT in the adluminal side transferred from the luminal side of the cultured blood–brain barrier (BBB) [80] (Figure 6). It was revealed that RAGE is required for transport of OT at the brain [80] and intestine [94], confirming that peripheral administration of OT may be useful in treatment of ASD or other psychiatry patients.

## 12. Conclusions and Perspective

The present review showed several neuronal roles for CD38 and CD157, in addition to their known functions in the digestive and immune systems. These two antigens are related to the neurodevelopment of neurons, astrocytes, and oligodendrocytes, and their impairment leads to multiple neurological and psychiatric conditions, such as ASD, schizophrenia, anorexia nervosa, suicidal ideation, and Parkinson’s disease. One core symptom of such diseases involves social impairments, some of which were replicated at the mouse level, in *Cd38* KO and/or *Cd157* KO mice, as useful animal models. CD38 and/or CD157 can be referred to as a neuro-entero-immunological regulator [6,18]. Interestingly, even for CD38, it is reported that the specific SNP (rs3796863) of *CD38* was originally documented as an association with ASD [21], with close relation to gastrointestinal dysfunction, one of the typical symptoms of ASD [95].

The low expression of CD38 in the embryonic brain raises questions associated with the early onset of ASD. In sharp contrast, the low level of CD157 expression in the adult brain raises another uncertainty in the late onset of Parkinson’s disease. Thus, future questions remaining involve how CD38 and CD157 play their roles in the embryonic or aging brain. Alternatively, we would like to know the more precise neurodysregulation by impaired CD38 and CD157 in the processes of neuronal development and neurodegeneration, which results in psychiatric disorders, such as ASD, schizophrenia and Alzheimer’s disease.

A recent topic is the cellular location of CD38 and association proteins, such as CAMKII, as shown in this article. Lee and Shao discovered other associated molecules or restricted amino acids in CD38 which can guide them to the topologically correct position as type II or type III glycoproteins [96]. In addition, it was found that sterile alpha and Toll/interleukin-1 receptor motif-containing 1 (SARM1), which is located in neuronal mitochondria and is activated by nicotinamide mononucleotide, displays enzymatic ability to form cADPR and ADPR from NAD^+^ [96,97], which is nearly identical to the enzyme activity of CD38. However, comparing cADPR for the OT release mechanism, it is considered to induce cell death (degeneration) in cortical neurons and axons [98,99], which is a pathophysiological aspect. This NAD^+^-related field will be studied intensively in near future.

Finally, the discovery of the OT-binding and transporter protein, RAGE [80,94], will bridge novel functional roles in the peripheral and central presence of OT. For more than 100 years since the discovery of OT, it is firmly thought that peripheral OT does not go into the brain, except for its information which is transmitted by the ascending sensory nerve. The discovery of the OT transporter not only contributes to solving the enigma of OT in clinical trials for ASD patients, but also adds a new concept for the brain function of this important hormone in social behavior.

## Figures and Tables

**Figure 1 cells-09-00062-f001:**
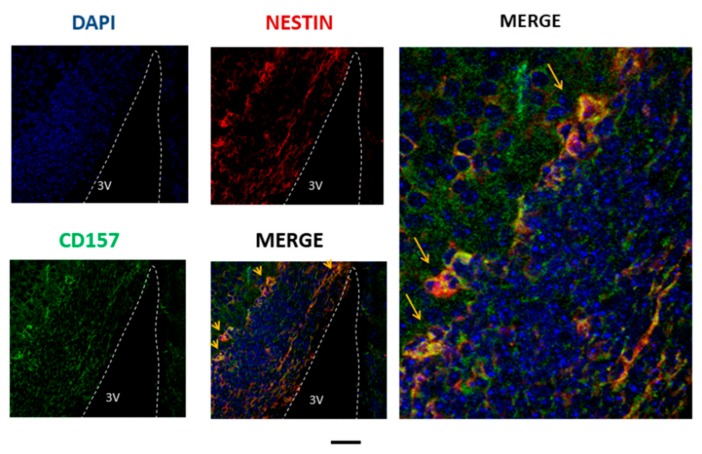
Cluster of differentiation 157 (CD157) expression analyzed by immunofluorescence staining in the embryonic mouse brain. Representative images were obtained from the E17 embryonic hypothalamus near the third ventricle (3V). CD157 is stained in green, nestin in red, and the nucleus in blue. Two merged images show the expression of CD157 in neural stem cells. Scale bar, 100 and 30 μm for the four left and enlarged images, respectively.

**Figure 2 cells-09-00062-f002:**
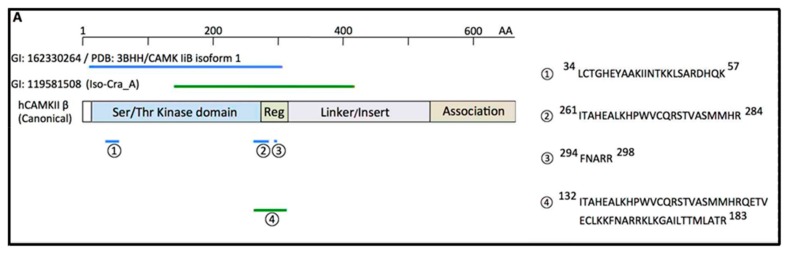
Identification of Ca^2+^-calmodulin-dependent protein kinase II (CAMKIIβ) as the human CD38 (hCD38)-associated protein by MALDI-TOF. The domain structures of the canonical sequence of human CAMKIIβ (isoform-4: UniProtKB/Swiss-Prot, Q13554) are shown as squares in the middle. The kinase domain (14–272) is shown in blue. The regulatory tail (273–315), linker region/insertions, and the association domain (532–664) are also depicted. The corresponding portions of “Chain A, crystal structure of CAMKIIβ isoform 1” (CAMK IiB isoform 1) and isoform CRA_A are marked as blue and green bars, respectively. The numbers 1–3 and 4 indicate the domains of CAMKIIβ at which CD38 binds, and sequences of four tryptic peptides of CAMKIIβ identified by MALDI-TOF/MS, respectively.

**Figure 3 cells-09-00062-f003:**
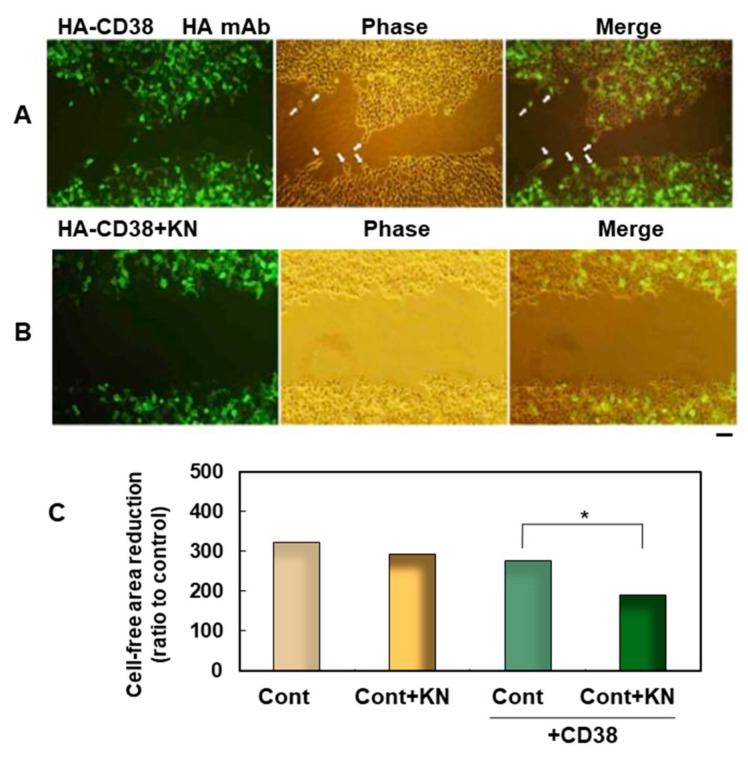
Effect of a CAMKII inhibitor on wound-induced migration facilitated by the expression of human influenza hemagglutinin (HA)-hCD38. (**A**,**B**) Human embryonic kidney 293T (HEK293T) cells grown on plastic dishes were transfected with *HA-hCD38* complementary DNA (cDNA) constructs. One day after transfection, a cell-free area was formed by scratching the plastic dishes, and they were treated with 0.1% dimethyl sulfoxide (DMSO) solution (control, A) or with 1 µM KN-62, a CaM kinase inhibitor, dissolved in 0.1% DMSO solution (**B**). At 24 h after scratching, they were fixed, permeabilized, and stained with HA monoclonal antibody (mAb). The right panels are merged images of fluorescent (Left) and phase-contrast (Middle) pictures. Scale bar, 100 µm. (**C**) Quantification of cell migration using the monolayer wound healing assay. The degree of migration was expressed as a reduction of the cell-free area by occupied cells. Each bar corresponds to 293T cells transfected with (green bars) or without (yellow bars) 1 µM of KN-62 (+KN). Each bar is the mean ± standard error of the mean (SEM) in seven experiments. Two-way ANOVA, *F*_3,7_ = 7.98, *p* = 0.0001. A significant difference was seen in the conditions of *CD38* transfection and KN treatment, but no significant interaction between transfection of *CD38* and treatment with KN-62 was observed (*p* = 0.18). Bonferroni’s post hoc analysis revealed a significant difference from the KN-free control value (**p* < 0.05).

**Figure 4 cells-09-00062-f004:**
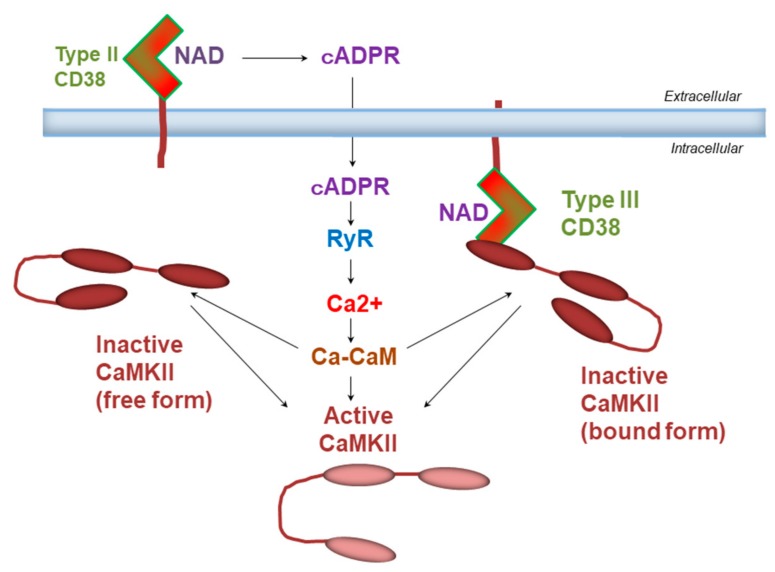
Hypotheses of CD38 and CaMKII interaction. Two models represent the possible means of activation of CaMKII by binding of calmodulin (CaM) and Ca^2^^+^ to the regulatory domain of CaMKII. Ca^2^^+^ is increased from ryanodine receptors bound by cyclic ADP-ribose (cADPR). cADPR is produced by either type II or type III CD38.

**Figure 5 cells-09-00062-f005:**
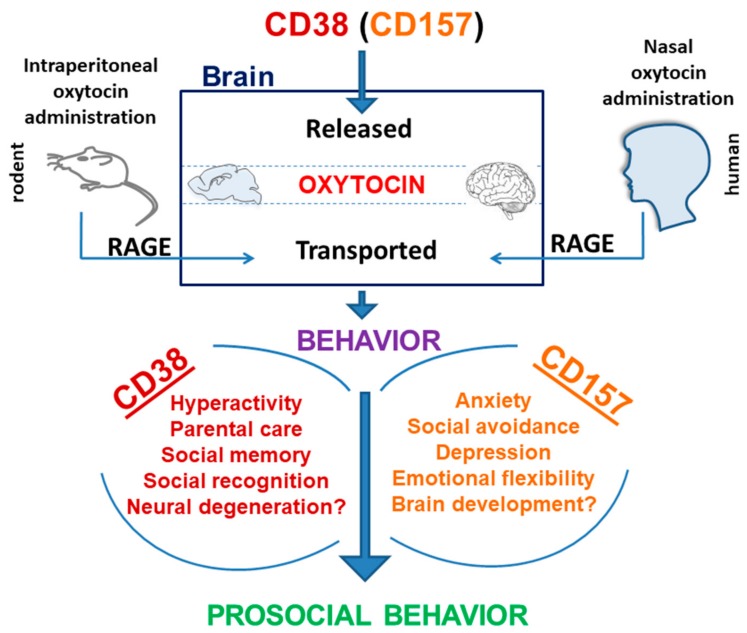
Scheme of CD38- and CD157-mediated pathways related to social behavior. The scheme shows the possible molecular mechanisms for increases in brain oxytocin (OT). One is OT release due to CD38 and CD157. Another is owing to receptor for advanced glycation end-products (RAGE)-dependent transport of OT into the brain. CD38- and CD157-dependent social behaviors are listed. The increased OT may trigger prosocial behavior.

**Figure 6 cells-09-00062-f006:**
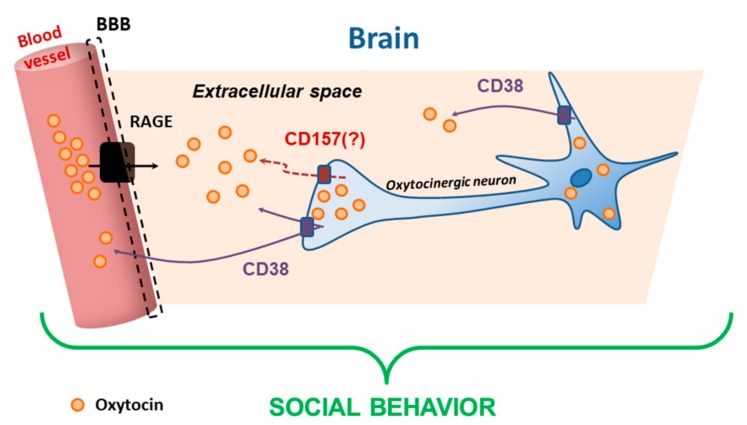
Scheme of the recruitment of OT in the brain extracellular space. OT in the blood is transported by RAGE on the endothelial cells of the blood–brain barrier (BBB). The OT in oxytocinergic neurons is secreted into the extracellular space via CD38 or CD157 from dendrites and axon collaterals. At the axon terminals, OT is released into the portal vein at the posterior pituitary. The turnover of OT may be a basis of social behavior.

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
