# Peer review of "CD38, CD157, and RAGE as Molecular Determinants for Social Behavior"

_cells, 2019, doi:10.3390/cells9010062_

Round 1

Reviewer 1 Report

Your review points to numerous areas requiring follow-up and further elucidations. The increasing prevalence of the Autism Spectrum Disroder and the great morbility carried out by Parkinson's Disease justifies additional studies, and although you briefly mention Schyzophrenia, the hypothesis of the neuronal role of CD38 and CD157 in impaired neuronal development raises de question of their effect in multiple neurological and psychiatric conditions. Both antigens have been reported to be associated to other processes including infection (HIV/AIDS, CD38) and immunoregulation (CD157). 

This reviewer feels a comment on the above situations would be complementary in your text, 

Author Response

We appreciated these thoughtful comments. Accordingly, we amended in Abstract and Conclusion as follow:

(Abstract) Single nucleotide polymorphisms of the CD38 and CD157/BST1 genes are associated with multiple neurological and psychiatric conditions, including autism spectrum disorder, Parkinson’s disease, and schizophrenia. In addition, both antigens are related to infectious and immunoregulational processes.

(Conclusion)      The present review show several neuronal roles for CD38 and CD157 in addition to their known functions in the digestive and immune systems. These two antigens are related to neurodevelopment of neurons, astrocytes, and oligodendrocytes and their impairment leads to multiple neurological and psychiatric conditions, such as ASD, schizophrenia, anorexia nervosa, suicidal ideation, and Parkinson’s disease. One of core symptoms of such diseases are social impairments, some of which were replicated at the mouse level, not all, but in Cd38 KO and/or Cd157 KO mice, as useful animal models.

Reviewer 2 Report

Higashida et al. reviewed the mechanisms by which CD38 and CD157 affect social behavior. Both CD38 and CD157 are immunologically related molecules and also have partially similar amino acid sequences. Some past studies have shown that these two genes determine social behavior. However, the expression of these two genes is different between embryos and adults. It is quite interesting to review the study of these genes in regulating social behavior. However, in this review, there are still many issues that need to be resolved. My comment is as follows:

When discussing SNPs in point 2, the subject of each paragraph is unclear, causing reading difficulties. The conclusion of point 2 requires clear conclusions to assist the reader in understanding the relevance of these SNPs to the disease.

In point 3, the authors cite a study by Jin et al. in 2007. The authors indicate that CD38 mRNA is expressed in the four brain regions of mice (hypothalamus, cerebellum, striatum, and cerebrum), but the expression levels in this paper are not clearly stated. Can the author provide more specific information?

3.

Line 104-105 “with no significant difference in density among them”. The meaning of this sentence is not clear, can you clearly describe the meaning of this sentence?

Line 148: Because in this paragraph, the authors show some of the features of CD157, but these do not match the title. This title needs to be re-edited.

The nuclear staining of Figure 1 is unclear and the location of the lineage cells cannot be confirmed. In addition, in order to more clearly observe the distribution of cells after staining, a wide range of tissue staining pictures should be added to show the distribution status.

The first paragraph in point 8 should give a clearer idea of ​​the meaning of TRPM2. The role of TRPM2 and temperature changes should be introduced first in the role of social behavioral disorders. At the same time, the association of TRPM2 with CD38/CD157 also requires a more logical description.

The title of point 9 refers to the social behavior of CD38 KO or CD157 KO mice. However, the social behavior of CD38 KO mice has not been described in detail. Please modify the title or increase the results of the CD38 KO mouse study.

Figure 5 presents a possible way for CD38 and CD157 to influence social behavior, but point 10 explores the role of CD157 that still needs further resolution. These issues should be presented in figure 5.

Author Response

When discussing SNPs in point 2, the subject of each paragraph is unclear, causing reading difficulties. The conclusion of point 2 requires clear conclusions to assist the reader in understanding the relevance of these SNPs to the disease.

We appreciated these suggestions. We amended the chapter as follow:

Genes and single nucleotide polymorphisms

The CD38 and CD157/BST1 genes locate on the subregion of the human chromosome 4p15 as a next neighbor. The genescape was well documented [3-7,19-21].

For an association study of CD38 and ASD, 10 intronic SNPs of CD38 were examined in a case-control study in a Japanese population. No significant association with ASD was identified in these SNPs [21]. Furthermore, when performed in the U.S. ASD DNA cohort (selected Caucasian 252 trios in the Autism Genetic Resource Exchange (AGRE) samples), none of selected SNPs showed significant associations [21]. However, if focused only in the U.S. high functioning autism subgroup, SNPs of 104 trios in our AGRE revealed association in rs6449197 (p=0.040) and rs3796863 (p=0.005). Unfortunately, no association was detected in Japanese high functioning autism trio cases (p=0.228). With respect to one exonic SNP, rs1800561 (4693C>T), some Asian ASD patients and controls possess arginine (dominant) and/or tryptophan of the 140th amino acid of CD38. Though there was no clear association in the SNP, ASD probands carrying tryptophan CD38, instead of arginine, was segregated in 3 Japanese families examined [21].

These initial SNP analyse for CD38 including rs3796863 were extended to infant’s attention to social eye cues [22] and replicated in ASD cases [23-28]. Most recently, in association studies of CD38, meaningful association was found in anorexia nervosa [29] and in suicidal ideation [26,30].

GWAS and meta-analyses for Parkinson’s disease identified intronic SNPs in the CD157/BST1 gene as new susceptibility locuses in Asian and European populations [13,14,31]. However, it was pointed out that environmental factors may also contribute in the real pathogenic role of CD157 SNPs on Parkinson’s disease [32].

Yokoyama et al. found associations between ASD and three SNPs of CD157 (rs4301112, rs28532698, and rs10001565) [33]. These three SNPs have chromosomal locations (from chr4:15717226 to chr14:15722573), being not identical from those associated with the region in Parkinson’s disease (chr14:15725766 to chr4:15737937) [13,14]. These studies revealed some SNPs in CD157 may be risk factors for both ASD and Parkinson’s disease and those in CD38 for several psychiatric disorders.

In point 3, the authors cite a study by Jin et al. in 2007. The authors indicate that CD38 mRNA is expressed in the four brain regions of mice (hypothalamus, cerebellum, striatum, and cerebrum), but the expression levels in this paper are not clearly stated. Can the author provide more specific information? Line 104-105 “with no significant difference in density among them”. The meaning of this sentence is not clear, can you clearly describe the meaning of this sentence?

However, such relation has not been studied yet in humans. Cd38 mRNA was expressed in the four mouse brain sub-regions (hypothalamus, cerebellum, striatum and cerebrum) [9], with the highest density in the hypothalamus (Supplementary Figure 11 of Jin et al. [9]).

Line 148: Because in this paragraph, the authors show some of the features of CD157, but these do not match the title. This title needs to be re-edited.

 According to this, we amended the subtitle as follow:

Immunohistochemistry of CD157 in neural stem cells

The nuclear staining of Figure 1 is unclear and the location of the lineage cells cannot be confirmed. In addition, in order to more clearly observe the distribution of cells after staining, a wide range of tissue staining pictures should be added to show the distribution status.

 It is very reasonable that we cannot identify in such small montages. We added the enlarged image in new Figure 1.

The first paragraph in point 8 should give a clearer idea of ​​the meaning of TRPM2. The role of TRPM2 and temperature changes should be introduced first in the role of social behavioral disorders. At the same time, the association of TRPM2 with CD38/CD157 also requires a more logical description.

We appreciated the kind comment. To avoid duplication from the previous publication, we deleted such explanation. But now we revived the story for TRPM2 in new words as follow:

Contribution of TRPM2 on Ca signaling

It is reported that NAD+ metabolites target several ion channels, including transient receptor potential melastatin 2 (TRPM2), which is a member of the warmth-sensing family plays an important role in thermoregulation [68,69]. Activation of TRPM2 non-specific cation channels results in Ca2+ influx in response to the temperature range from 34 to 40°C [68]. Therefore, initially, we thought that OT release could be facilitated by activation of TRPM2 channels. Extracellular application of 100 mM cADPR increases intracellular free calcium concentrations, together with simultaneous temperature elevation by 2°C from 35°C in the presence of extracellular Ca2+ in a single cultured cell of the anterior hypothalamus of the mouse [68].

The title of point 9 refers to the social behavior of CD38 KO or CD157 KO mice. However, the social behavior of CD38 KO mice has not been described in detail. Please modify the title or increase the results of the CD38 KO mouse study.

 Folloing the suggestion , we amended as follow:

Social behavior in Cd157 mice

Figure 5 presents a possible way for CD38 and CD157 to influence social behavior, but point 10 explores the role of CD157 that still needs further resolution. These issues should be presented in figure 5.

Thank you   the quite right comment. We agree with this. So we added two phrases for the roles of CD157 and CD38. These are Neural degeneration? and Brain development?

Reviewer 3 Report

The review, "CD38, CD157 and RAGE as molecular determinants for social behavior" is a very interesting study with a significant implication in signaling changes in brain which is behavioral alteration. The study could be very important for drug target finding in Autism or Parkinson's or other related disorders.

The references for each section is sufficient to establish the importance of that particular section to connect with CD38/CD157 dependent hypothesis. They cover evidences from gene to mRNA to protein and further association of these with cell line, mouse/animal and human. The conclusion part is very short and not concise. That part need to be more well written to combine all the studies they mentioned here. 

Author Response

Thank you very much for these reasonable comments. Following your suggestion we remake conclusion as follow:

Conclusions and perspective

                  The present review show several neuronal roles for CD38 and CD157 in addition to their known functions in the digestive and immune systems. These two antigens are related to neurodevelopment of neurons, astrocytes, and oligodendrocytes and their impairment leads to multiple neurological and psychiatric conditions, such as ASD, schizophrenia, anorexia nervosa, suicidal ideation, and Parkinson’s disease. One of core symptoms of such diseases are social impairments, some of which were replicated at the mouse level, not all, but in Cd38 KO and/or Cd157 KO mice, as useful animal models. CD38 and/or CD157 can be referred to as a neuro-entero-immunological regulator [6,18]. Interestingly, even for CD38, it is reported that the specific SNP (rs3796863) of CD38 is originally documented an association with ASD [21] has close relation with gastrointestinal dysfunction, one of typical symptoms of ASD [96].

               The low expression of CD38 in the embryonic brain raises questions associated with the early onset of ASD. In sharp contrast, the low level of CD157 expression in the adult brain raises another uncertainty in the late onset of Parkinson’s disease. Thus, future questions remained are how CD38 and CD157 play their roles in the embryonic or aging brain. Alternatively saying, we would like to know the more precise neurodysregulation by impaired CD38 and CD157 in the processes of neuronal development and neurodegeneration which results in psychiatric disorders, as ASD, schizophrenia and Alzheimer’s disease.

               One of recent topics is the cellular location of CD38 and association proteins, such as CAMKII, as shown in this article. Lee and Shao discovered other associated molecules or restricted amino acids in CD38 which can guide to the topologically right position as type II or type III glycoproteins [97]. In addition, it is found that sterile alpha and Toll/interleukin-1 receptor motif-containing 1 (SARM1), which locates in neuronal mitochondria and is activated by nicotinamide mononucleotide, display enzymatic ability to form cADPR and ADPR from NAD+ [97, 98], which is nearly identical enzyme activities of CD38. However, comparing cADPR for the OT release mechanism, it is considered to induce cell death (degeneration) in cortical neurons and axons [98], pathophysiological aspects. This NAD+ related field will be studied intensively in near future.

            Finally, the discovery of the OT binding and transporter protein, RAGE [80,95], will bridge novel functional roles in peripheral and central presence of OT. For more than 100 years since the discovery of OT [99], it is firmly thought that peripheral OT does not go into the brain, except for its information which is transmitted by the ascending sensory nerve. The discovery of the OT transporter not only contributes to solving the enigma of OT in clinical trials for ASD patients, but also adds a new concept for the brain function of this important hormone in the social behavior.